# Superior Mesenteric Artery Occlusion Caused by Infective Endocarditis and Worsened by Mycotic Aneurysm and Intracranial Hemorrhage: A Case Report

**DOI:** 10.3390/medicina58111585

**Published:** 2022-11-03

**Authors:** Changho Kim, Tak-Hyuk Oh

**Affiliations:** 1Department of Emergency Medicine, School of Medicine, Kyungpook National University, 680, Gukchaebosang-ro, Jung-gu, Daegu 41944, Korea; 2Department of Thoracic & Cardiovascular Surgery, Kyungpook National University Chilgok Hospital, Daegu 41404, Korea

**Keywords:** superior mesenteric artery, endocarditis, mycotic aneurysm, intracranial hemorrhage

## Abstract

The superior mesenteric artery (SMA) is more commonly occluded than other abdominal arteries due to anatomical factors. Though rare, SMA occlusion is life-threatening. We present the case of a 50-year-old male patient who presented with fever and abdominal pain and was subsequently diagnosed with SMA embolism, SMA mycotic aneurysm, and infective endocarditis. Many patients visit the emergency room complaining of abdominal pain. Although SMA occlusion diagnosis is rare in these cases, detailed examination and close monitoring of patients are warranted considering the high mortality rate of this disease.

## 1. Introduction

Superior mesenteric artery (SMA) occlusion is a life-threatening disease that causes severe abdominal pain and has a mortality rate of 74%. The condition is rare, accounting for 0.09–0.2% of all acute surgical admissions, and can be caused by several factors, including infective endocarditis of the mitral valve (MV), albeit rarely [1,2]. In this report, the patient complained of abdominal pain and was admitted for SMA occlusion. The patient did not have a history of atherosclerosis, and in the causal examination, infective endocarditis with mitral valve vegetation was diagnosed. Embolic occlusive mesenteric ischemia, which accounts for 50% of acute mesenteric ischemia, was determined and treatment was initiated. The patient was diagnosed with SMA embolic occlusion and underwent heart valve surgery and treatment with drugs. However, the patient’s condition worsened with the development of intracranial hemorrhage, a presumed sequela of anticoagulation treatment, and SMA mycotic aneurysm. After an additional heart valve surgery and surgical treatment for SMA aneurysms (SMAA), the patient was followed up without issue.

## 2. Case Report

A 50-year-old male patient visited the emergency medical center with fever and abdominal pain for 20 days. The patient was transferred to our hospital after an abdominal computed tomography (CT) scan performed at another hospital showed liver and spleen infarctions. Splenic infarction and SMA occlusion were diagnosed after a detailed examination. Initially, a conservative treatment plan without surgery was decided on. As a side effect of anticoagulant treatment, the patient developed intracranial hemorrhage (ICH). Additionally, infective endocarditis with vegetation was diagnosed using echocardiography, and MV replacement (MVR) was performed at the thoracic surgery clinic. Thereafter, he complained of abdominal pain and fever and was diagnosed with an SMA mycotic aneurysm following an abdominal CT. An aneurysm resection treatment was subsequently performed, and the patient was later treated with antibiotics and anticoagulants.

## 3. Investigations

The first blood test performed at the emergency medical center showed the following results: C-reactive protein level, 7.74 mg/dL (≤0.30 mg/dL); white blood cell count, 21.03 × 10^3^/uL (3.91–10.33 × 10^3^/uL); hemoglobin level, 10.8 g/dL (11.9–15.4 g/dL); D-dimer level, 2.74 ug/mL (0.00–0.50 ug/mL); and troponin I (Tn-I) level, 0.148 ng/mL (≤0.034 ng/mL). Blood cultures found Streptococcus gordonii, and a CT scan showed splenic infarction and SMA obstruction (Figure 1). In addition, echocardiography performed with heat showed MV anterior leaflet echogenic material (1.00 × 0.38 cm), MV posterior leaflet oscillating echogenic material (1.99 *×* 0.66 cm), and moderate to severe mitral valve regurgitation (Figure 2). MVR (Figure 3) was performed in thoracic surgery. Moreover, the patient began complaining of headache after the initiation of anticoagulation treatment, and ICH was diagnosed using a brain CT. The patient’s abdominal pain and fever persisted during subsequent follow-ups, despite treatment. An additional abdominal CT showed a low-density lesion (12 mm) with an enhancing wall in the SMA, and surgery was performed based on the diagnosis of SMA mycotic aneurysm. The treatment plan, examinations, and diagnoses required interdepartmental cooperation with specialized departments such as emergency medicine, gastroenterology, transplant and vascular surgery, thoracic surgery, radiology, neurosurgery, and cardiology.

## 4. Treatment

The patient visited the emergency medical center and was initially diagnosed with SMA occlusion during examination. There was no evidence of peritonitis; therefore, no surgical treatment was required. The patient was started on systemic anticoagulation with intravenous heparin, which was subsequently changed to oral warfarin. After echocardiography, infective endocarditis with vegetation was confirmed as the cause of the persistent fever. An emergency thoracotomy was performed, and an MVR was conducted with the placement of a 31–33 mm On-X mechanical valve. A CT scan performed eight days after cardiac surgery showed an SMA embolism located 7 cm distal to the SMA origin and distal to the takeoff of the middle colic artery and the first few jejunal branches. The thrombosed SMA showed aneurysmal change and the diameter had increased to 19 × 15 mm (Figure 4). Upon surgical exploration the day following the CT scan, the aneurysm sac revealed a purulent thrombus and necrotic wall, confirming infectious etiology. We performed an aneurysmectomy of the SMA without any vascular reconstruction due to the color and motility of the small bowel being intact (Figure 5). Due to their rarity and cryptic presentations, infected aneurysms in the SMA must be diagnosed with a high index of suspicion. Early blood cultures and an investigation into the origins of the infection are crucial. Surgical excision should be considered, given that a residual infected aneurysm may cause sepsis, spread, or rupture.

## 5. Outcome and Follow Up

One week after aneurysm resection surgery, the patient was on warfarin with an international normalized ratio between 1.1 and 2.7. The use of antibiotics for infective endocarditis was continued.

## 6. Discussion

### 6.1. The Patient Visited the Hospital with Abdominal Pain and Fever, and SMA Occlusion Was Diagnosed—The Patient Was Initially Observed without Surgery

Acute mesenteric ischemia (AMI) is a sudden interruption of blood supply to visceral organs, and if untreated may lead to ischemia, cellular damage, intestinal necrosis, and death. Mesenteric ischemia is a rare condition that accounts for 0.09–0.2% of emergency surgeries and is associated with high morbidity and mortality, if not diligently addressed [1]. SMA embolic occlusion is the most common cause, due to anatomical factors, including the low takeoff angle and wide diameter of the SMA. Arterial emboli are the most frequent cause of AMI and are responsible for approximately 40–50% of cases [2]. Emboli may occur in the left atrium (LA) of the heart due to arrhythmias, such as atrial fibrillation, which presents approximately 3–10 cm distal to the origin of the SMA. Additionally, approximately 20% of concurrent emboli form in the spleen and kidney [1]. Preoperative systemic anticoagulation is required in the treatment of SMA embolization to prevent the propagation of clots around the embolus as well as to guard against further embolization of the intestines or other organs. After arrival to the emergency room, a CT scan of the patient showed segmental occlusion of the SMA with no definite evidence of bowel ischemia and multiple hypodense areas in the spleen (Figure 1).

### 6.2. Infective Endocarditis Was Found on Examination and Thoracic Surgery Was Performed

Blood tests of the patient showed an elevated CRP level (from 7.74 mg/dL to 17.67 mg/dL), bacteremia (peripheral blood culture: Streptococcus gordonii), and an elevated Tn-I level. In the 2D-echo examination of the heart, infectious endocarditis, MV vegetation, and echogenic material (1.00 × 0.38 cm) were observed in the MV anterior leaflet (1.00 × 0.38 cm). Oscillating echogenic material (1.99 × 0.66 cm) was also observed in the MV posterior leaflet (Figure 2). Most mesenteric emboli originate from cardiac sources. However, the literature on mesenteric ischemia in patients with infective endocarditis is limited, with only a few published case reports available [3]. A recent review of the published literature concluded that vegetations sized >10 mm are a strong indication for surgery [4]. Moderate to severe mitral valve regurgitation and vegetation of the heart were diagnosed in our case. Therefore, MVR was performed, and a 31–33 mm On-X mechanical valve was surgically placed during the emergency thoracotomy in the thoracic surgery department (Figure 2). Antibiotic and anticoagulation (warfarin) treatments were subsequently initiated. The first consideration, in this case, was whether early cardiac valve surgery may have prevented this complication. The size and mobility of vegetations are strong independent predictors of a new embolic event. Particular microorganisms, including staphylococci, Streptococcus bovis, and Candida species, may also increase the risk of embolization, but are of secondary importance. The patient in our case was diagnosed with infective endocarditis after admission, and blood cultures showed Streptococcus gordonii. Ceftriaxone and gentamicin were administered inpatient, and an oral third-generation cephalosporin was prescribed upon discharge. The European Society of Cardiology guidelines state that there is no indication for the initiation of antithrombotic drugs, including thrombolytic drugs, anticoagulants, or antiplatelet therapy, during the active phase of infective endocarditis [5].

### 6.3. The Patient Was Treated with Anticoagulants—During Hospitalization, He Complained of Severe Headache without Neurological Symptoms, and a Brain CT Scan Showed Intracranial Hemorrhage

The patient was postoperatively treated with antibiotics and anticoagulants. During inpatient follow-ups, the patient complained of a severe headache; therefore, a brain CT scan was obtained (Figure 3). Patients may experience this as a side effect of other anticoagulant treatments. However, no clear guidelines are available in the literature to support this observation. An increased risk of cerebral hemorrhage in patients with infective endocarditis who were taking oral anticoagulants was reported in a few studies. However, the European Society of Cardiology guidelines do not provide specific recommendations for infective endocarditis cases. Neurological symptoms were not present in the patient, and the cerebral hemorrhage was minor; therefore, a conservative treatment plan was followed.

### 6.4. The Patient Was Diagnosed with SMA Aneurysm Using Abdominal CT, Which Was Followed by Persistent Abdominal Pain and Fever—Surgery Was Performed on the Transplanted Vessel

SMAAs are rare, with 5.5% occurring due to visceral artery aneurysms. The condition is characterized by the expansion and eventual rupture of the aneurysm, which affects 38–50% of patients with SMAA [6]. Owing to its rarity, diagnosis of SMAA may be difficult, and the aforementioned signs and symptoms are often mistaken for more frequent conditions, such as pancreatitis, perforated viscus, ulcer disease, or appendicitis [6]. To date, duplex scan ultrasonography has been considered a valuable, quick, and reliable diagnostic tool. However, most SMAAs are currently diagnosed preoperatively using computed tomography angiography (CTA) or angiography [7]. Significant advances in three-dimensional and multiplanar imaging software have made it possible to obtain high-resolution images of the abdominal aorta and its branches; therefore, CTA is the gold standard technique for this type of aneurysm [8]. The patient was diagnosed with an SMA mycotic aneurysm using abdominal CTA following persistent abdominal pain and inflammatory findings (Figure 4). Surgery for aneurysm resection and SMA mycotic aneurysm was performed during vascular surgery (Figure 5). After the follow-up, small bowel and colon motility showed normal clinical symptoms. The patient did not exhibit any other unusual fever or abdominal pain after surgical treatment. Typically, the decision to pursue vascular reconstruction depends primarily on the patient’s underlying vascular status, which determines the likelihood of ischemia distal to the site following aneurysm excision and the anatomical site of the aneurysm [9].

### 6.5. A Multidisciplinary Approach, Involving Emergency Medicine, Radiology, Thoracic Surgery, Transplant and Vascular Surgery, and Neurosurgery Specialists Was Required

The patient came to our hospital with abdominal pain and was diagnosed with SMA occlusion, infective endocarditis, brain hemorrhage, and SMA mycotic aneurysm, and confirming the initial examination was essential. CRP and leukocyte levels, which are indicators of inflammation, were elevated, and d-dimer levels were also elevated. For this reason, CT and echocardiography were performed to identify various potential causes of SMA occlusion, which ultimately led to the diagnosis of infective endocarditis. In addition, diagnosing SMA mycotic aneurysm by CT scan after hospitalization was possible, showing persistent CRP and WBC elevations and abdominal pain. After two surgical treatments, the patient was hospitalized and received antibiotics and anticoagulants. A multidisciplinary approach was crucial for treatment, involving specialists in emergency medicine, radiology, thoracic surgery, vascular surgery, and neurosurgery. Despite the high mortality rate associated with this condition, better patient outcomes may be achieved through multidisciplinary cooperation. In addition, controversy regarding intracranial hemorrhage as a side effect of anticoagulation treatment exists. Therefore, the decision on its usage must be at the discretion of the medical staff after a detailed analysis of the situation.

## 7. Conclusions

SMA occlusion is a rare disease. However, it should be considered as a cause in patients with severe abdominal pain if other causes are ruled out.Examining the cause of SMA occlusion is necessary, and infective endocarditis should be suspected when accompanied by fever.For infective endocarditis with vegetation over 10 mm, rapid valve surgery should be considered.Anticoagulation treatment in this disease is controversial, and side effects such as intracranial hemorrhage should always be considered.

## Figures and Tables

**Figure 1 medicina-58-01585-f001:**
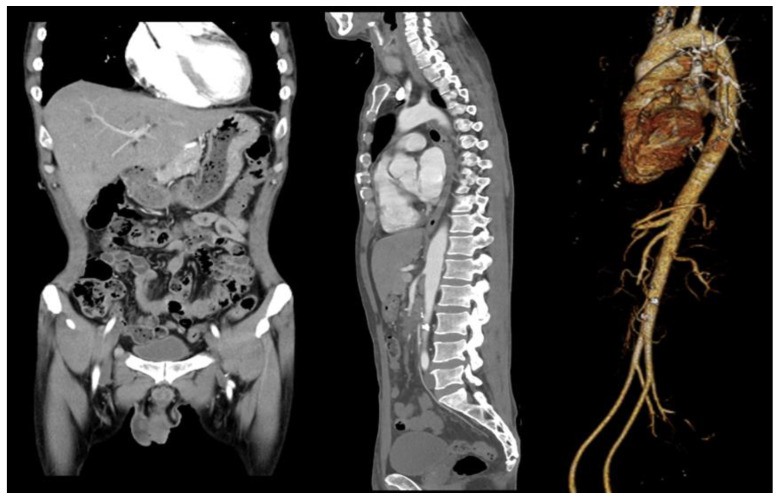
Initial computed tomography (CT) scanning performed in the emergency room showing superior mesenteric artery (SMA) embolism (coronal, sagittal, angio).

**Figure 2 medicina-58-01585-f002:**
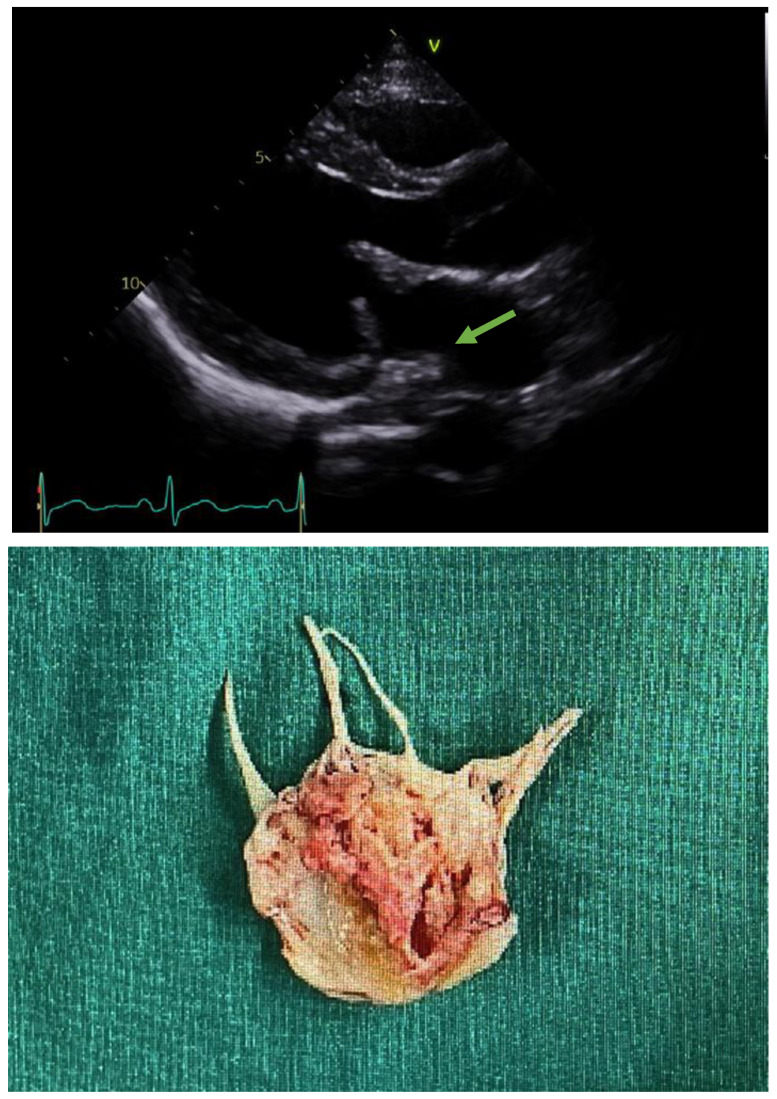
Infective endocarditis vegetation. The first image shows the echocardiography after hospitalization, while the second image is post-cardiothoracic surgery.

**Figure 3 medicina-58-01585-f003:**
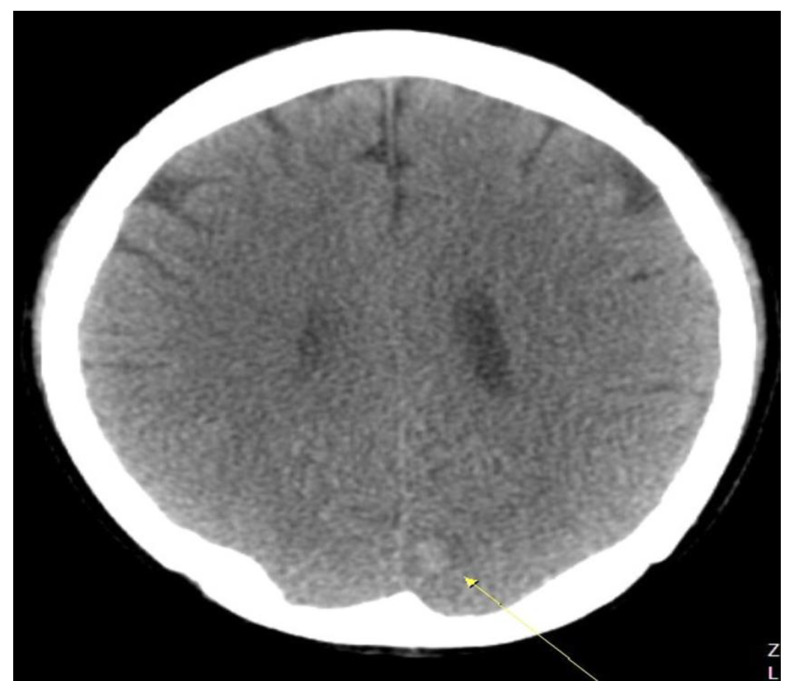
Brain CT scan showing intracranial hemorrhage three days after admission.

**Figure 4 medicina-58-01585-f004:**
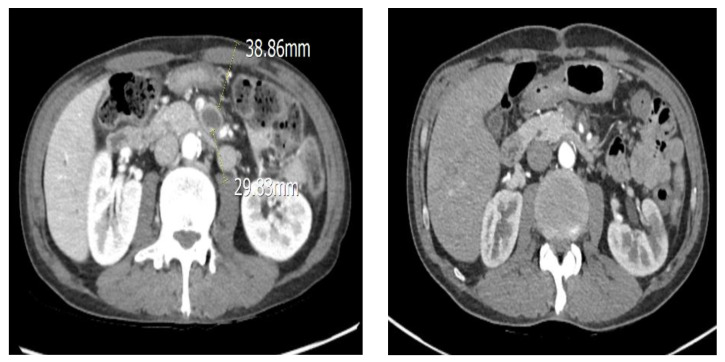
CT showing SMA mycotic aneurysm. The first picture is pre-operative, while the second is post-operative.

**Figure 5 medicina-58-01585-f005:**
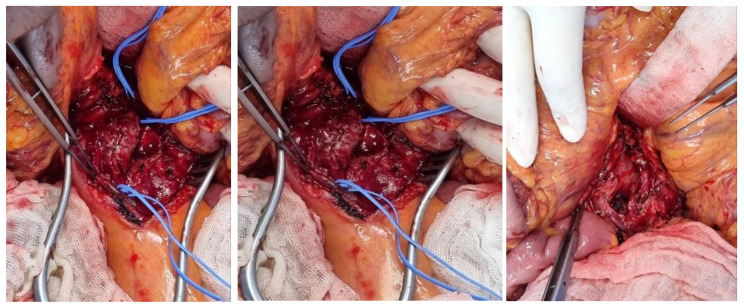
SMA mycotic aneurysm operation room finding. The first two images are prior to excision with pus, and the last image is after the excision.

## Data Availability

Not applicable.

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
