# Peer review of "Superior Mesenteric Artery Occlusion Caused by Infective Endocarditis and Worsened by Mycotic Aneurysm and Intracranial Hemorrhage: A Case Report"

_medicina, 2022, doi:10.3390/medicina58111585_

Round 1

Reviewer 1 Report

This is an interesting case and it adds knowledge to the medical literature. However, I have the following issues: 

1. More information on the resection of the superior mesenteric artery aneurysm is needed. How was the artery reconstructed? 

2. Can the authors give a post-operative control CT angiography showing the state of the mesenteric artery? 

3. It would be very interesting to see the evolution of splenic infarction. 

4. How long was the patient's progression tracked? And how it was done? 

5. Was imaging control performed after discharge?

6. Figure captions should be more comprehensive. 

Also, the overall appearance of the manuscript could be improved.

Author Response

Dear reviewer

Thank you so much for the review on my case thesis. I've tried to proofread the thesis as much as possible as you advised, and if you need more information, please let me know.
Thanks again, and I will try my best to write a better thesis.

Reviewer 2 Report

The authors presented a case report outlining the importance of the superior mesenteric artery occlusion caused by infective endocarditis, in the pathological context. Also it was highlighted the importance of properly treated infective endocarditis for the disease outcome While the manuscript has several strengths, there are few issues that need to be addressed:

1.      In “Introduction” a paragraph with the relation between SMA and atherosclerosis / thrombosis would be appropriate. Also, a medical history of the patient, mostly related to his cardiovascular profile is very important for the overview.

2.      In “Investigations” I would recommend to add to the clinical description a complete table of clinical parameters, each accompanied by normal vales in brackets.

3.      In “Treatment” specify which antibiotics was used.

4.      In “Discussion” it must be referred also to other clinical parameters important for the case, besides CRP, like number of white blood cells, d-dimer levels etc

5.      In “Figures” a complete figure legend is needed, specify both the investigation used and the result obtained (figure description).

Author Response

(The authors gave the same response as above.)
